# Repurposing Niclosamide to Modulate Renal RNA-Binding Protein HuR for the Treatment of Diabetic Nephropathy in db/db Mice

**DOI:** 10.3390/ijms25179651

**Published:** 2024-09-06

**Authors:** Lili Zhuang, Wenjin Liu, Xiao-Qing Tsai, Connor Outtrim, Anna Tang, Zhou Wang, Yufeng Huang

**Affiliations:** Division of Nephrology & Hypertension, Department of Internal Medicine, University of Utah Health, Salt Lake City, UT 84132, USA; lili.zhuang@hsc.utah.edu (L.Z.); liuwj1989@hotmail.com (W.L.); xtsai@alumni.nd.edu (X.-Q.T.); connorouttrimceo@gmail.com (C.O.); 05anna.tang@gmail.com (A.T.); zhou.wang@hs.utah.edu (Z.W.)

**Keywords:** RNA-binding protein, HuR inhibitor, diabetic kidney disease, renal inflammation, oxidative stress

## Abstract

Hu antigen R (HuR) plays a key role in regulating genes critical to the pathogenesis of diabetic nephropathy (DN). This study investigates the therapeutic potential of niclosamide (NCS) as an HuR inhibitor in DN. Uninephrectomized mice were assigned to four groups: normal control; untreated db/db mice terminated at 14 and 22 weeks, respectively; and db/db mice treated with NCS (20 mg/kg daily via i.p.) from weeks 18 to 22. Increased HuR expression was observed in diabetic kidneys from db/db mice, which was mitigated by NCS treatment. Untreated db/db mice exhibited obesity, progressive hyperglycemia, albuminuria, kidney hypertrophy and glomerular mesangial matrix expansion, increased renal production of fibronectin and a-smooth muscle actin, and decreased glomerular WT-1^+^-podocytes and nephrin expression. NCS treatment did not affect mouse body weight, but reduced blood glucose and HbA1c levels and halted the DN progression observed in untreated db/db mice. Renal production of inflammatory and oxidative stress markers (NF-κBp65, TNF-a, MCP-1) and urine MDA levels increased during disease progression in db/db mice but were halted by NCS treatment. Additionally, the Wnt1-signaling-pathway downstream factor, Wisp1, was identified as a key downstream mediator of HuR-dependent action and found to be markedly increased in db/db mouse kidneys, which was normalized by NCS treatment. These findings suggest that inhibition of HuR with NCS is therapeutic for DN by improving hyperglycemia, renal inflammation, and oxidative stress. The reduction in renal Wisp1 expression also contributes to its renoprotective effects. This study supports the potential of repurposing HuR inhibitors as a novel therapy for DN.

## 1. Introduction

Diabetic kidney disease (DKD) is the leading cause of end-stage renal disease (ESRD) in the US and is increasing worldwide. DKD is fundamentally a biochemical process with complex metabolic patterns that result in oxidative stress and chronic inflammation, both of which play crucial roles in the pathogenesis and progression of DKD. It has been established that angiotensin-converting enzyme inhibitors (ACEi), angiotensin receptor blockers (ARBs), and current sodium-glucose type 2 transporter inhibitors (SGLT2i) can reduce disease progression. However, the majority of patients with DKD still progress to ESRD, highlighting the need for novel disease-modifying therapeutic targets, such as directly targeting oxidative stress and/or inflammation in DKD.

Hu antigen R (HuR), also known as embryonic lethal abnormal vision-like protein (ELAVL1), is a ubiquitously expressed post-transcriptional regulator [1]. It binds to adenine- and uridine-rich elements (AREs) located in the 3′-untranslated region (3′-UTR) of mRNA in response to various stimuli, facilitating mRNA transport from the nucleus to the cytoplasm and preventing rapid degradation, thereby enhancing the generation and action of these molecules [2]. Notably, many pro-inflammatory and key profibrotic transcripts contain conserved or semi-conserved AREs in their 3′-UTR [3]. Elevated HuR, which regulates molecules involved in organ inflammation and fibrosis, has been observed in many diseases, including chronic kidney disease (CKD) with or without diabetes [4,5,6,7,8,9], especially in kidney biopsy tissues from patients with DKD [10,11,12]. The consistent upregulation of HuR contributes to persistent inflammation, playing a pivotal role in kidney disease. Importantly, current drugs for treating DKD, such as ACEi, ARB, or SGLT2i, do not impact elevated HuR expression and action in DKD in animal models (unpublished data).

The discovery of specific small-molecule HuR antagonists, such as KH3 and KH39, which selectively disrupt the HuR–ARE interaction and inhibit downstream protein expression, has shown substantial effectiveness in the progression of CKD and cardiovascular disease (CVD) in various animal models [10,11,13]. Consequently, therapeutics targeting HuR may offer a unique and promising approach for the treatment of DKD that extends beyond the benefits achieved with ACEi/ARBs or SGLT2i.

Although specific HuR inhibitors are still in preclinical development, some FDA-approved anthelminthic drugs, such as pyrvinium pamoate (PP), have been reported to inhibit HuR by preventing its cytoplasmic accumulation [14]. Niclosamide (NCS) is more tolerable and safer than PP as an anthelminthic drug [15] and has also been identified as an effective HuR inhibitor in preclinical studies [16,17]. Our observations further show that NCS significantly inhibits cellular HuR expression and its cytoplasmic translocation and improves renal function, albuminuria, renal inflammation and fibrosis in a mouse model of septic kidney injury induced by repeated lipopolysaccharide (LPS) injections (manuscript under review). Based on these promising findings, we hypothesize that NCS-mediated HuR inhibition will downregulate critical genes involved in renal inflammation and fibrosis, potentially leading to remission of DKD. 

NCS has a proven safety profile from its use in treating tapeworm infections and has emerged as a promising candidate in drug repurposing efforts due to its multifunctional properties observed in multiple animal models. Research has shown that NCS can modulate several key proinflammatory signaling pathways, such as Wnt/ß-catenin, mTORC1, STATs, NF-kB, Notch and NS2B–NS3 interaction, in addition to its anthelmintic effects [18,19,20,21]. Despite its potential, the effectiveness of NCS in clinical trials, including those for cancer treatment, remains uncertain, primarily due to its limited bioavailability, and its long-term impacts on various diseases are still under investigation. Recent studies have reported that adding NCS to ACE inhibitor therapy in diabetic nephropathy patients resulted in further reductions in albuminuria [22]. However, it remains unclear if these effects are mediated through its action on HuR-dependent pathways, even though many of the affected molecules are known HuR targets. Therefore, in this study, we employed the uninephrectomized db/db mouse model, a well-established type 2 diabetes mellitus (T2DM) model [23,24,25], to assess the therapeutic potential of NCS and its influence on HuR-mediated mechanisms. To overcome its low oral bioavailability, NCS was dissolved in a special solvent and administered intraperitoneally to mice in this study. 

## 2. Results

### 2.1. HuR Is Increased in Diabetic Kidneys in db/db Mice

As shown in Figure 1A, HuR protein (stained red) was primarily localized in the nucleus of kidney cells in non-diabetic control mice. No cytoplasmic staining of HuR was observed in these normal kidney cells. In contrast, kidneys from diabetic db/db mice exhibited markedly increased overall HuR staining intensity in the nucleus of glomerular and tubular cells, along with noticeable cytoplasmic staining in glomeruli (as indicated by arrows). Treatment with NCS decreased the overall HuR staining intensity compared to untreated diabetic kidneys. Immunoblot analysis corroborated these findings, showing a significant increase in total HuR protein levels in diabetic mouse kidneys as early as 14 weeks of age, with further increases observed by 22 weeks compared to normal controls. NCS treatment significantly reduced these elevated HuR levels (Figure 1B,C). These results suggest that renal HuR expression is elevated in diabetic db/db mice and that NCS effectively inhibits this increase.

### 2.2. Treatment with NCS Reduces Hyperglycemia and Albuminuria

Table 1 presents the characteristics of the diabetic mice. The diabetic db/db mice had greater body weight than the non-diabetic controls, showing obesity. There was no significant difference in body weight between treated and untreated db/db mice, either at baseline or at the end of the study. The untreated db/db mice maintained high blood glucose levels with significantly elevated glycosylated hemoglobin (HbA1c) as the disease progressed. Consequently, these mice had notably increased daily water intake and urine output. NCS treatment slightly but significantly reduced hyperglycemia in db/db mice, leading to a clinically significant reduction in HbA1c levels from 13.3% to 10.8%, compared to untreated db/db mice measured at 22 weeks of age. This reduction in glucose levels likely contributed to the decreased daily urinary volume observed in the NCS-treated db/db mice. 

Plasma blood urea nitrogen (BUN) and creatinine (Cr) levels were higher in the diabetic db/db mice compared to the non-diabetic controls at both 14 and 22 weeks of age. NCS treatment significantly improved these renal function markers, with plasma BUN and creatinine levels falling below baseline levels observed at 14 weeks. These findings indicate that NCS treatment ameliorates impaired renal function in db/db mice.

Assessment of 24 h urinary albumin excretion (UAE) revealed that the diabetic db/db mice excreted much more albumin than the normal controls as early as 14 weeks of age. UAE levels continued to rise until 22 weeks of age, consistent with previous reports [24]. NCS treatment, starting at 18 weeks of age, effectively halted the progressive increase in albuminuria in the db/db mice (Figure 2). 

### 2.3. Treatment with NCS Reduces Kidney Hypertrophy, Fibrosis and Podocyte Injury

As expected, diabetic db/db mice showed continually increased kidney weight from 14 to 22 weeks, compared to non-diabetic controls, indicating diabetes-induced kidney hypertrophy. Surprisingly, treatment with NCS reversed this increase, bringing it closer to baseline levels (Table 1). This result was further confirmed by measuring glomerular size, which in the db/db mice at 14 weeks of age was 1.59-fold larger than in normal mice and further increased to 2.11-fold at 22 weeks. Treatment with NCS from 18 weeks to 22 weeks effectively inhibited the progressive enlargement of glomerular size (Figure 3C). At 14 weeks of age, the db/db mice exhibited increased glomerulosclerosis compared to the non-diabetic controls, characterized by substantial accumulation of PAS-positive pink extracellular matrix (ECM) proteins and enhanced deposition of matrix protein collagen IV (Col-IV) in the glomerular mesangium. This accumulation of ECM and Col-IV intensified further by 22 weeks of age (Figure 3A–E). NCS treatment reduced the progressive ECM accumulation in diabetic glomeruli by 57.8% and effectively reversed the increased glomerular deposition of Col-IV in diabetic db/db mice.

We also evaluated the protein levels of fibronectin (FN) and a-smooth muscle actin (a-SMA), key markers of fibrosis, using Western blot analysis of renal cortex tissue (Figure 3F–H). Consistent with the increased glomerulosclerosis, the levels of these fibrotic markers were significantly elevated from weeks 14 to 22 in db/db mice. NCS treatment completely reversed the renal deposition of FN and a-SMA, bringing their levels closer to those observed in the non-diabetic controls.

Additionally, renal mRNA levels of FN, collagen type I (Col-I), and Col-IV were increased in the db/db mice compared to the normal controls (Figure 3I–K). NCS treatment significantly decreased these elevated mRNA levels. These findings collectively demonstrate that NCS treatment effectively halts the progression of diabetes-induced kidney hypertrophy and glomerulosclerosis in db/db mice. 

As shown in Figure 4A,B, immunofluorescent staining revealed intense nephrin staining, a marker for podocyte slit diaphragm [26], in the glomeruli of the non-diabetic controls. In the db/db mice, nephrin staining was reduced by 41% (*p* < 0.05) at 14 weeks, a level that persisted at 22 weeks. However, NCS treatment largely restored nephrin staining to levels comparable to those of the non-diabetic controls.

Wilms’ tumor gene 1 (WT-1), a nuclear protein specific to podocytes [27,28], was used to count podocytes within the glomerular area (Figure 4A,C). Consistent with the nephrin staining results, glomeruli from the db/db mice had fewer podocytes compared to the non-diabetic controls. The NCS-treated db/db mice, however, exhibited an increased number of podocytes compared to untreated db/db mice.

Overall, these data suggest that NCS treatment ameliorates podocyte dysfunction in diabetic db/db mice, as evidenced by reduced podocyte loss and preserved expression of slit-diagram proteins. This improvement may contribute to a significant reduction in albuminuria and remission of glomerulosclerosis. 

### 2.4. Treatment with NCS Reduces Inflammation and Oxidative Stress Involved in the Progression of Diabetic Nephropathy

To assess the renoprotective effect of NCS on proinflammatory pathways, we evaluated the protein production of NF-kB-p65 and NAPDH oxidase-2 (Nox2) in the renal cortex of the diabetic db/db mice. Both NF-kBp65 and Nox2 were significantly elevated in these mice compared to the non-diabetic controls (Figure 5), indicating substantial activation of these pathways in diabetic kidney tissue.

In line with these changes, urine malondialdehyde (MDA) levels, an indicator of renal oxidative stress, were markedly higher in db/db mice compared to normal controls (Table 2). Treatment with NCS effectively reduced these oxidative stress markers to near-normal levels. Additionally, increased renal NF-kB-p65 production and elevated urinary levels of TNF-a and MCP-1 observed in db/db mice were significantly decreased following NCS treatment (Figure 5 and Table 2). 

In total, these results suggest that NCS treatment effectively mitigates renal inflammation mediated by NF-kB and oxidative stress driven by NAPDH oxidase. 

### 2.5. Treatment with NCS Improves the Renal Imbalance of Angiopoietin (Angpt) 1 and 2 Involved in Diabetes-Induced Endothelial Dysfunction

Microvascular dysfunction, particularly injury to the capillary endothelium, is a critical factor in the pathogenesis of DN [29]. This dysfunction can contribute to glomerular hypertrophy and increased protein leakage through the endothelium, potentially exacerbating podocyte injury. Angiopoietins and their receptor play important roles in maintaining microvascular endothelium hemostasis [30]. In diabetic kidneys, the decreased secretion of Angpt1 or a reduced Angpt1 to Angpt2 ratio often leads to reduced Angpt1-Tie2 signaling, which is central to DN pathophysiology [31]. We further investigated how NCS treatment affects renal production of Angpt1 and Angpt2.

As anticipated, the diabetic db/db mice showed a marked decrease in Angpt1 and an increase in Angpt2 protein levels in kidney tissue at 14 weeks, with this imbalance worsening by 22 weeks (Figure 6). Surprisingly, NCS treatment reversed the diabetes-induced reduction in Angpt1 levels, although it had only a minor effect on the elevated Angpt2 levels. These findings suggest that NCS therapy may improve renal endothelial function, primarily through the upregulation of Angpt1 and Angpt1’s signaling pathways. 

### 2.6. Treatment with NCS Downregulates Renal WNT1-Inducible-Signaling Pathway Protein 1 (Wisp1) in db/db Mice

Wisp1 has been shown to drive both inflammatory and fibrotic responses, contributing to the development of renal fibrosis and progression of CKD. Of note, it was recently identified as a key downstream mediator of HuR-dependent action [32]. We thus investigated whether NCS treatment affects Wisp1’s generation and action in db/db mice. As shown in Figure 7, both Wisp1mRNA and protein expression levels were significantly increased in the 22-week-old diabetic kidney tissues, but not in 14-week-old tissues. These increases were markedly reversed by NCS treatment, bringing them to near normal levels. The effect of NCS on Wisp1 is consistent with the inhibitory action of NCS on renal HuR, thereby contributing to its renoprotective effect in db/db mice.

## 3. Discussion

The db/db mouse is a hyper-insulinemic, genetically diabetic model with obesity that exhibits renal abnormalities akin to those seen in human nephropathy associated with type 2 diabetes mellitus (T2DM). Our previous research demonstrated that uninephrectomy at 8 weeks of age significantly accelerates DN development in db/db mice. These mice showed elevated albuminuria starting at 10 weeks and progressed to advanced DN with increasing albuminuria and glomerulosclerosis by 22 weeks of age [24]. In the present study, we evaluated the therapeutic effect of NCS in progressive DN using this uninephrectomized db/db mouse model. To investigate whether NCS treatment could slow, stop, or even reverse the progression of DN, we selected two disease progression stages: from weeks 14 to 22, and weeks 18 to 22. However, we only treated the diabetic db/db mice from 18 to 22 weeks. Surprisingly, treatment with NCS not only significantly inhibited elevated HuR expression and its key downstream target Wisp1 in diabetic kidneys, but also ameliorated progressive albuminuria, podocyte injury, renal endothelial function and glomerulosclerosis, as well as related inflammatory and oxidative stress pathways. Most of these measurements after NCS treatment were close to, or even better than, the baseline levels determined in the db/db mice at 14 weeks of age, indicating that NCS treatment halts or even partially reverses the progression of DN in db/db mice. NCS treatment also reduces glycated HbA1c levels by 2.48%, consistent with a previous study in db/db mice [33], which demonstrates its efficacy in improving hyperglycemia in T2DM. Thus, the effect of NCS in preventing the progression of DN in db/db mice is likely related to its impact on HuR-dependent pathways and glucose levels. 

Although we did not specify kidney cell types with diabetes-induced elevated HuR, multiple cells may be involved in diabetic kidneys. It has been shown that hyperglycemia-induced elevated HuR in renal podocytes, mesangial cells, tubular cells and DKD mice is linked to podocyte, mesangial cell, and tubular cell injury, as well as inflammation and oxidative stress, thereby promoting the progression of DN [5,34,35]. Clinically, elevated renal histological HuR levels have been confirmed as an independent risk factor for disease progression in DN patients [12]. Therefore, the inhibitory effect of NCS on renal HuR expression and function in db/db mice may be involved in ameliorating kidney cell injury associated with diabetes, thereby leading to the improvements in renal functional and histological outcomes observed in the present study. 

To date, there are no reports on hyperglycemia-induced elevated HuR specifically in the microvasculature in the kidney. However, microvascular dysfunction, particularly damage to the capillary endothelium, remains the primary cause of DN, analogous to its role in diabetic retinopathy [29]. In diabetic retinopathy, HuR overexpression has been shown to promote angiogenesis by stabilizing VEGF-A expression and influencing the angiogenic activity of endothelial cells (ECs) [36]. Furthermore, HuR inhibition via siRNA has been shown effective in counteracting diabetic retinopathy in animal models [37]. In the present study, we observed that NCS treatment improved renal Angpt1 production and the Angpt1 to Angpt2 ratio in diabetic kidneys, suggesting a potential benefit in renal endothelial integrity and function. Whether this improvement in endothelial function results directly from NCS’s effect on HuR in glomerular endothelial cells requires further investigation. Nonetheless, the observed reduction in podocyte injury may contribute to the enhanced Angpt1 production and improved endothelial function in db/db mice, as noted in our previous work [31]. 

It has been well-established that inflammation and oxidative damage, driven by elevated glucose levels in diabetes, significantly contribute to the development and progression of diabetic complications [38,39]. Addressing these pathways, in addition to managing glucose level, has emerged as a promising therapeutic approach for diabetic vascular complications, including DN [40,41,42]. Increasing evidence indicates that elevated HuR expression and its nucleocytoplasmic translocation can stabilize itself, as well as inflammatory mediators and NAPDH oxidases, in response to various stimuli, thereby amplifying inflammatory and oxidative stress signals [10,11]. This stabilization occurs because many pro-inflammatory transcripts, NADPH oxidases, and HuR itself contain AU-rich elements in their 3′ untranslated regions [3,34,35]. The enhanced activity of HuR and its interaction with its targets likely establish an HuR/pro-inflammatory and oxidative circuit, which may perpetuate inflammation and oxidative damage across various organs. Therefore, targeting HuR could provide an effective strategy for mitigating renal inflammation and oxidative stress in diabetes. Previous studies, along with ours, have shown that inhibiting HuR significantly reduces NF-kB signaling, nucleotide-binding oligomerization domain 2 (NOD2) or IL-17-mediated inflammation, and Nox2- or Nox4-mediated oxidative stress in kidney diseases [4,10,34,35]. In the present study, NCS treatment reduced elevated renal inflammatory and oxidative stress markers, including NF-kB, TNF-a, MCP-1, Nox2 and MDA levels. NCS is known for its multifunctional anti-inflammatory properties by inhibiting the Wnt/ß-catenin-, mTORC1-, STATS-, NF-kb-, Notch- and NS2B-NS3-mediated inflammation [18,19,20,21]. However, it has not yet determined whether the broad effects of NCS on these molecular pathways result from its action on HuR-dependent regulation, despite many of the affected molecules being HuR targets. Given the specific inhibitory effect of NCS on HuR expression and nucleocytoplasmic translocation in multiple cells in response to various injuries and in diabetic kidneys, the impact of NCS on both inflammation and oxidative stress observed in db/db mice is likely mediated through multiple HuR-targeted pathways. 

Wisp1, a member of the connective tissue growth factor and nephroblastoma overexpressed families, plays a significant role in differentiation, proliferation, ECM remodeling and NF-kB-mediated inflammation across various cell types [43]. Wisp1 expression is upregulated in tumor cells as well as in response to stress or injury conditions in the kidney, such as ischemia-reperfusion injury, unilateral ureter obstruction, or diabetes [44,45,46,47]. Recently, it has been shown that HuR directly binds Wisp1 mRNA and regulates its expression in myofibroblasts in response to fibrotic stimuli in a TGFß-dependent manner [32]. Our preliminary findings indicate that NCS treatment also inhibits Wisp1 overexpression in diabetic kidneys, further supporting the direct inhibitory effect of NCS on HuR and its downstream targets. The reduction in Wisp1 and its associated signaling pathways may contribute to the amelioration of DN. 

Interestingly, NCS has been shown to lower blood glucose levels in various animal models of diabetes, including STZ-induced type 1 diabetic mice, type 2 diabetic db/db mice and high-fat-diet-induced type 2 diabetic mice [33,48,49,50,51]. This glucose-lowering effect is thought to result from improvements in insulin signaling, reduced insulin resistance, decreased hepatic glucose production, and increased energy expenditure. Consistent with these findings, our study observed that NCS treatment significantly reduced both glucose levels and HbA1c levels in diabetic db/db mice, indicating improved blood glucose control. However, the study did not delve into the underlying mechanisms of these effects. It is likely that the observed improvements in renal HuR expression and the Angpt1/Angpt2 ratio with NCS treatment also apply to the pancreas in db/db mice. In addition to its known effects on glucose levels, this new action of NCS may contribute to the amelioration of pancreatic islet inflammation and oxidative stress, potentially leading to enhanced islet ß-cell function and better glucose control. 

Together, NCS shows therapeutic potential for DN through its modulation of multiple targets, primarily by affecting the common post-transcriptional regulator HuR. 

In summary, NCS effectively inhibits HuR in diabetic kidneys and shows significant potential for ameliorating hyperglycemia, reducing renal inflammation and oxidative stress, and inhibiting the progression of albuminuria and glomerulosclerosis in type 2 diabetic db/db mice. These findings provide strong proof-of-principle for repurposing NCS as an HuR inhibitor, potentially offering a promising adjunctive therapy alongside current treatments to prevent renal failure associated with diabetes. The observed therapeutic reduction in albuminuria with NCS may also explain its benefits in diabetic nephropathy patients [22]. However, despite NCS’s long history of safe use treating tapeworm infections without apparent side effects, further investigation is needed to assess its safety and efficacy for long-term use in DKD or other CKD patients. Additionally, addressing the current low bioavailability of NCS is key to enhancing its effectiveness in these patients.

## 4. Materials and Methods

### 4.1. Reagents

Unless specified, all reagents were obtained from Sigma Chemical (St. Louis, MO, USA). 

### 4.2. Animals and Experimental Designs

Thirty-five male mice, comprising eight non-diabetic C57bl/6 and twenty-seven db/db mice at 6 weeks of age, were obtained from Jackson Laboratory (Bar Harbor, ME, USA). All mice underwent right nephrectomy under anesthesia at 8 weeks of age to accelerate the development of diabetic nephropathy, following established protocols [23,24,25]. At 14 weeks of age, baseline measurements were taken for all mice, including body weight (BW), water intake, urine output, blood glucose (BG), blood HbA1c, and urinary albumin (UAE). Based on these baseline measurements, twenty-seven male uninephrectomized diabetic db/db mice were randomized into 3 groups as follows: group 1, non-diabetic normal control; group 2, diabetic db/db mice without treatment, terminated at 14 weeks, serving as a baseline control; group 3, diabetic db/db mice without treatment, terminated at 22 weeks, serving as disease progression control; and group 4, diabetic db/db mice treated with niclosamide (NCS, 20 mg/kg body weight) daily from 18 weeks to 22 weeks of age. To overcome its low oral bioavailability, NCS (catalogue No. 481909) at a concentration of 1mg/mL was dissolved in PBS containing 5% ethanol and 5% Tween-80. The solution was prepared freshly with sonication and administered intraperitoneally to mice on a daily base. Details regarding the preparation and dosing of NCS have been described in previous studies [16,17]. All mice were housed in standard cages with a 12 h light/dark cycle, given water and normal diet ad libitum, and euthanized under isoflurane anesthesia as scheduled. Body weight, blood glucose level, and urinary albumin excretion were monitored every other week. Mice in group 2 were euthanized at 14 weeks of age to establish the baseline disease severity, while all the other mice were euthanized at 22 weeks of age. 

### 4.3. Euthanasia

All mice were euthanized under isoflurane anesthesia at the specified time points. Blood samples were collected via heart puncture. Plasma samples were prepared for the measurement of blood urea nitrogen (BUN) and creatinine (Cr) levels. All tissues were perfused with 40 mL to 60 mL of ice-cold sterilized phosphate-buffered saline (PBS) through the heart. The perfused kidney was removed. Renal cortical and medullar tissue were separated. Portions of renal cortex tissue were either snap-frozen in 2-methylbutane at −80 °C, fixed in 10% neutralized formalin for immunohistology examination, or stored in liquid nitrogen for protein isolation for Western blot analysis. 

The animal maintenance and study procedures described herein were conducted at the University of Utah, adhering to the Public Health Service Policy on Use of Laboratory Animals and approved by the Institutional Animal Care and Use Committee (IACUC) at the University of Utah.

### 4.4. Measurement of Clinical Parameters

The blood glucose level and glycosylated hemoglobin (Hb_A1C_) level were monitored using tail blood samples. Blood glucose was measured with a Glucometer Elite XL (Bayer Healthcare, Elkhart, IN, USA), and HbA1c was assessed using the DC 2000+ HbA1c kit (catalogue No. 5068A, Bayer Healthcare). Plasma BUN and Cr levels were determined by using the QuantiChromTM urea assay kit (catalogue No. DIUR-100, Bio-Assay System, Hayward, CA, USA) and creatinine assay kit (catalogue No. 80350, Crystal Chem Inc. Elk Grove Village, IL, USA).

Twenty-four-hour urine collections were made from each mouse using a metabolic cage at 14 weeks of age, prior to treatment, and at the time of sacrifice. Urine albumin levels were measured using the DC2000+ microalbumin reagent kit (catalogue No. 6011A, Bayer Healthcare), as previously described [23]. Urine levels of monocyte chemoattractant protein-1 (MCP-1) and tumor necrosis factor-alpha (TNF-a) were quantified using the commercially available ELISA kits (catalogue No. 88-7391-88 and No. 88-7324-88, Invitrogen, Carlsbad, CA, USA). Urine malondialdehyde (MDA), also known as thibarbituric acid-reactive substances (TBARS), was measured using a colorimetric assay (catalogue No. 10009055, Cayman Chemical Company, Ann Arbor, MI, USA). The levels of MCP-1, TNF-a or MDA excretion were expressed as the total amount excreted over 24 h.

### 4.5. Histological Analysis of Kidney Tissue

Formalin-fixed renal cortex tissues were subsequently embedded in paraffin, and 4 µm sections were cut from the kidney tissue blocks. The sections were stained with periodic acid Schiff (PAS) to assess glomerular size and extracellular matrix (ECM). Quantification was performed either directly using image-J (NIH-ImageJ 1.53t, Bethesda, MD, USA) to calculate the area of each glomerulus or as a PAS-positive pink area using a computer-assisted color image analysis system, as previously described [23,24,25]. Twenty glomeruli from each mouse were evaluated at magnification (×400). The percentage of mesangial matrix occupying each glomerulus was determined based on the PAS-positive material in the mesangium. The average ECM score was calculated by averaging scores from all glomeruli on a section per mouse. All analyses were conducted in a blinded manner by two individuals.

Immunofluorescent staining for HuR was performed on paraffin-embedded kidney tissues as previously described [10]. DAPI-Fluoromount-G (SouthernBiotech, Birmingham, AL, USA) was used to stain the nuclei DNA, indicating cellular position of HuR. Control slides treated with antibody diluent alone showed no staining. 

Immunofluorescent staining for type IV collagen (Col-IV), Wilms tumor protein-1 (WT-1) and nephrin was conducted on frozen kidney sections as previously described [9,52]. Intraglomerular Col-IV and nephrin staining was quantified by calculating the percentage of positive staining over the glomerular area in at least 15 randomly selected glomeruli per section, using image-J software (NIH-ImageJ 1.53t, Bethesda, MD, USA) [52,53]. The number of WT-1-positive podocytes per glomerulus was counted in 20 randomly selected glomeruli per section. Control slides treated only with PBS instead of primary antibodies showed no staining. 

### 4.6. Western Blot Analysis of Kidney Tissue

Western blot assays were conducted as previously described [23,24,25,31]. Briefly, 20 mg of renal cortex tissue from each mouse was homogenized in lysis buffer (Cell Signaling Technology, Beverly, MA, USA), supplemented with 1% NP-40, 1 mM PMSF, and a protease inhibitor mix (1 tablet/5 mL). Protein samples from each mouse in a group were pooled and subjected to SDS-PAGE in 4–12% gradient gel (Invitrogen, Carlsbad, CA, USA). The proteins were then transferred to immobilon-P membranes (Millipore, Bedford, MA, USA) and immunoblotted. The following proteins were assessed: fibronectin (FN), a-smooth muscle actin (a-SMA), NF-kB-p65, NAPDH oxidase-2 (Nox2), angiopoietin-1 (Angpt1), Angpt2, WNT1-inducible-signaling pathway protein 1 (Wisp1), ß-actin, and GAPDH. The antibodies used for the Western blot assays and immunofluorescent staining are listed in Appendix A. Quantitation of the bands was performed using a Bio-Rad Molecular Imager ChemiDoc^TM^ XRS+ system (Bio-Rad Laboratories, Inc., Hercules, CA, USA). Each protein level was normalized to the densitometric intensity of GAPDH or ß-actin. For comparison, this ratio for normal control samples was set to unity, and other lanes on the same gel were expressed as fold-changes relative to this value. All blots were run in triplicate.

### 4.7. RNA Isolation and Real-Time RT-PCR Assay

Total RNA was extracted from kidney cortex tissue using Tri Reagent (ThermoFisher Scientific, Waltham, MA, USA) following the manufacturer’s instructions. RNA transcription and q-PCR were carried out using the superscript III first-strand synthesis system (Invitrogen, Carlsbad, CA, USA) and the SYBR green dye I (Applied Biosystems, Foster City, CA, USA) as previously described [23]. Samples run as triplicates in separate tubes were analyzed using the 2^^(−∆∆Ct)^ method and quantified with results normalized to GAPDH. Relative mRNA levels of the target gene were expressed relative to the non-diabetic normal kidney control (NC), which was set at unity. The sequences of primers used for the targeted molecules are listed in Table 3. The PCR product was confirmed by 1.5% agarose gel electrophoresis, which showed a single band of the expected size.

### 4.8. Statistical Analysis

Data are expressed as mean ± SD. To accommodate the skewed distribution of urinary albumin, TNF-a, MCP-1, and MDA excretion values, these data were log10-transformed before statistical comparison. Differences among groups were analyzed using two-way ANOVA, following by Student–Newman–Keuls or Dunnett’s post hoc tests for multiple comparisons. A *p*-value of < 0.05 was considered statistically significant.

## Figures and Tables

**Figure 1 ijms-25-09651-f001:**
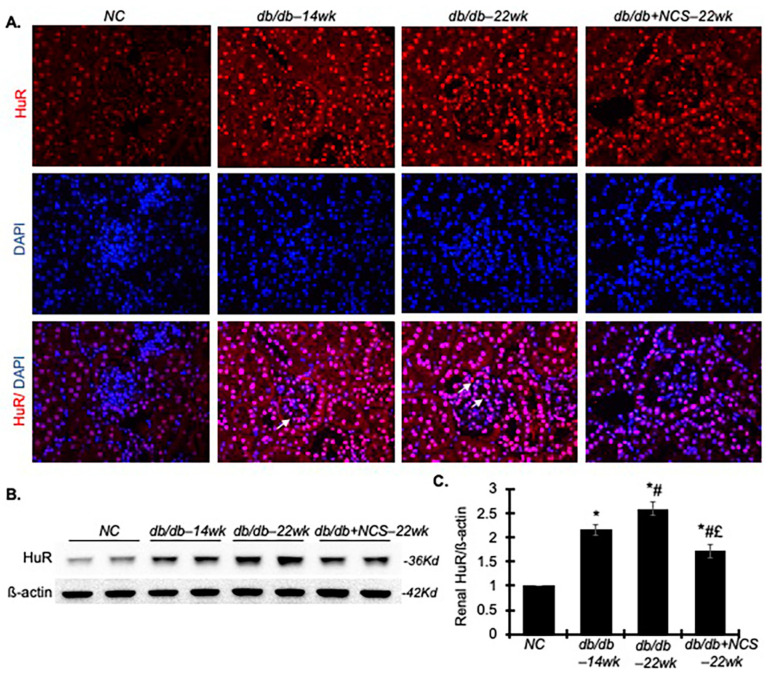
Increased renal HuR staining, and protein production were observed in the diabetic db/db mouse kidney. (**A**) Representative photomicrographs of renal immunofluorescent staining for HuR (red) at 400× magnification are shown from normal mice (NC), diabetic db/db mice at 14 weeks (db/db–14wk), diabetic db/db mice at 22 weeks (db/db–22wk) and diabetic db/db mice treated with NCS at 22 weeks (db/db + NCS–22wk). A few cells with cytoplasmic staining for HuR are indicated by arrows in the diabetic kidneys. (**B**) Representative Western blots illustrate the total cellular protein expression of HuR and ß-actin in the renal cortex tissue. (**C**) Quantification of the Western blot band density. Protein values are expressed as fold-changes relative to the normal control, which was set to unity. * *p* < 0.05, vs. NC; # *p* < 0.05, vs. db/db–14wk; £ *p* < 0.05, vs. db/db–22wk.

**Figure 2 ijms-25-09651-f002:**
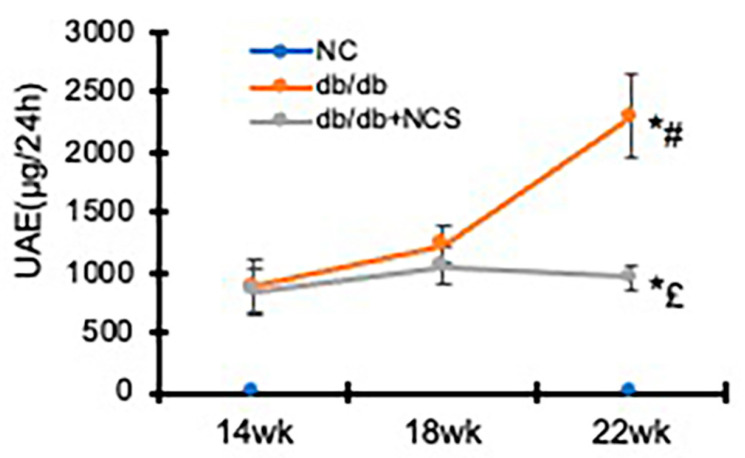
Treatment with NCS arrests the progression of albuminuria in diabetic db/db mice. Urine and urinary albumin excretion levels over 24 h (UAE/24 h) were collected and determined at the ages of 14, 18, and 22 weeks, as described in the Section 4. * *p* < 0.05, vs. NC; # *p* < 0.05, vs. db/db–14wk; £ *p* < 0.05, vs. db/db–22wk.

**Figure 3 ijms-25-09651-f003:**
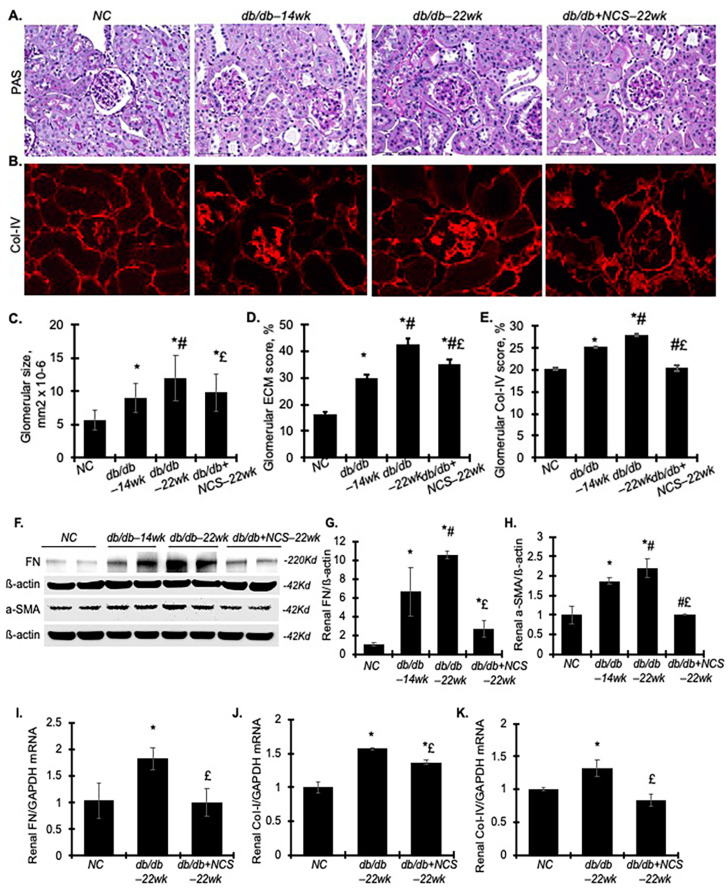
Treatment with NCS reduces glomerular hypertrophy, glomerulosclerosis, glomerular matrix protein deposition and expression in diabetic db/db mice. (**A**) The representative microscopic images illustrate PAS staining of kidney sections, which was used to detect glomerular size and extracellular matrix (ECM) deposition (stained pink). Magnification, ×400. (**B**) Representative photomicrographs of glomerular immunofluorescent staining for type IV collagen (Col-IV). Magnification, ×400. (**C**–**E**) The graphs summarize the results of average glomerular size (**C**), glomerular ECM deposition (**D**) and glomerular Col-IV staining score (**E**), quantified using image-J. (**F**) Western blots of FN, a-SMA, and ß-actin from normal mouse kidneys and diabetic kidneys of untreated and treated mice. Molecular weight is labelled on the right. (**G**,**H**) The graphs present the results of band density measurements for FN (**G**) and a-SMA (**H**) in the kidneys. The protein values are expressed relative to normal control, which was set to unity. (**I**–**K**) The graphs show the relative mRNA levels of FN (**I**), Collagen I-a1 (Col-I) (**J**), and Collagen IV-a1 (Col-IV) (**K**) in the kidneys, as determined by the real-time RT–PCR assay. * *p* < 0.05, vs. NC; # *p* < 0.05, vs. db/db–14wk; £ *p* < 0.05, vs. db/db–22wk.

**Figure 4 ijms-25-09651-f004:**
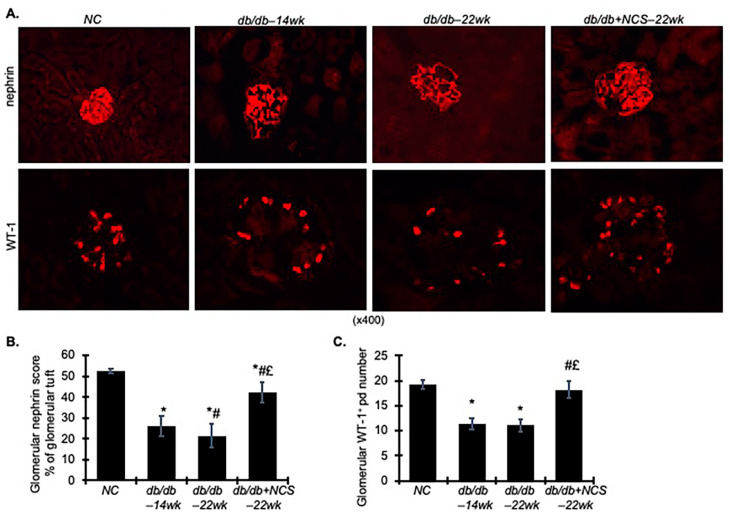
Treatment with NCS reverses the glomerular podocyte number and nephrin expression in diabetic db/db mice. (**A**) Kidney sections from normal mice (NC), diabetic db/db mice at 14 weeks (db/db–14wk), diabetic db/db mice at 22 weeks (db/db–22wk) and diabetic db/db mice treated with NCS at 22 weeks (db/db + NCS–22wk) were stained with nephrin and WT-1-postive podocytes. Magnification, 400×. (**B**,**C**) The graphs summarize the results of glomerular nephrin staining (**B**) and WT-1^+^ cells (**C**), quantified using image-J. * *p* < 0.05, vs. NC; # *p* < 0.05, vs. db/db–14wk; £ *p* < 0.05, vs. db/db–22wk.

**Figure 5 ijms-25-09651-f005:**
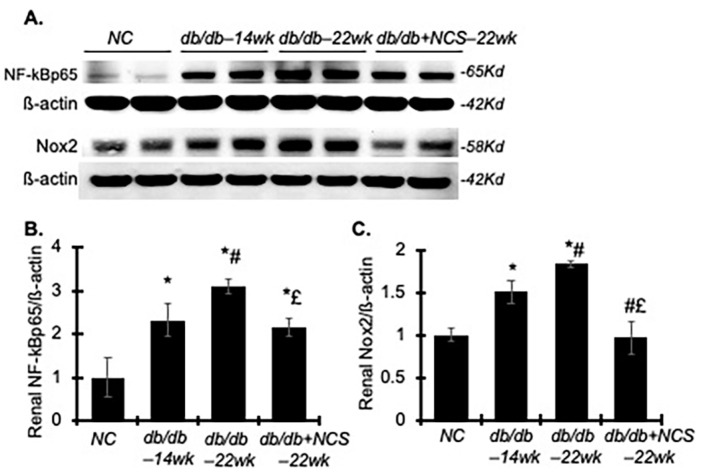
Treatment with NCS reduces renal NF-kBp65 and Nox2 protein production in diabetic db/db mice. (**A**) Representative Western blots illustrate the protein expression of NF-kBp65, Nox2 and ß-actin in the kidney tissue from normal mice and diabetic untreated and treated mice. Molecular weight is labelled on the right. (**B**,**C**) The graphs present the results of band density measurements for NF-kBp65 (**B**) and Nox2 (**C**) in the kidneys. Protein values are expressed relative to normal control, which was set to unity. * *p* < 0.05, vs. NC; # *p* < 0.05, vs. db/db–14wk; £ *p* < 0.05, vs. db/db–22wk.

**Figure 6 ijms-25-09651-f006:**
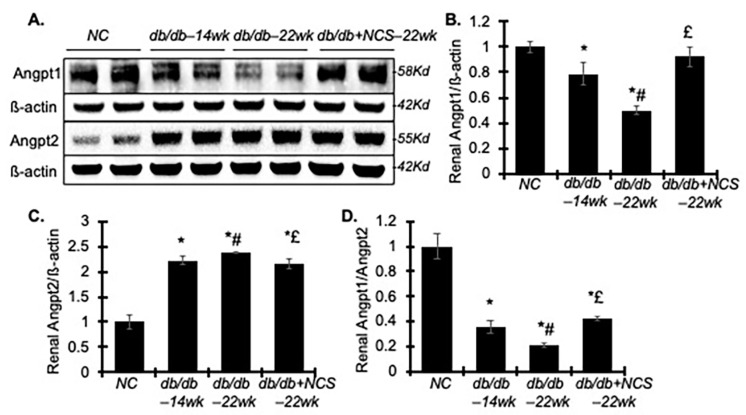
Treatment with NCS ameliorates renal angiopoietin (Angpt) 1 and 2 expression in diabetic db/db mice. (**A**) Representative Western blots illustrate the protein expression of Angpt1, Angpt2 and ß-actin in the kidney tissue from normal mice and diabetic untreated and treated mice. Molecular weight is labelled on the right. (**B**–**D**) The graphs present the results of band density measurements for Angpt1 (**B**), Angpt2 (**C**) and the ratio of Angpt1 to Angpt2 (**D**) in the kidneys. Protein values are expressed relative to normal control, which was set to unity. * *p* < 0.05, vs. NC; # *p* < 0.05, vs. db/db–14wk; £ *p* < 0.05, vs. db/db–22wk.

**Figure 7 ijms-25-09651-f007:**
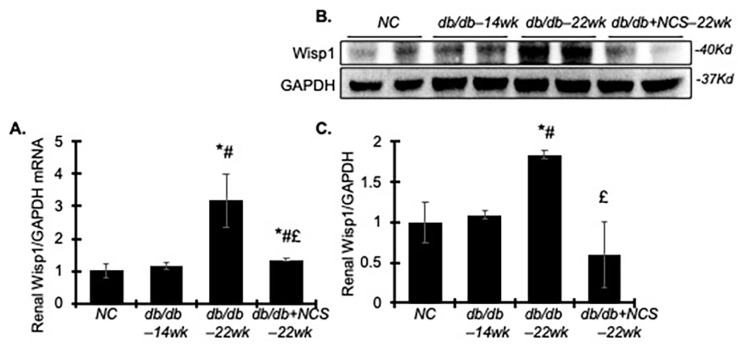
Treatment with NCS reduces renal mRNA and protein expression of Wisp1 in diabetic db/db mice. (**A**) The graph shows the relative mRNA levels of Wisp1 in the kidneys, as determined by the real-time RT–PCR assay. (**B**) Representative Western blots illustrate the protein expression of Wisp1 and GAPDH in the kidney tissue from normal mice and diabetic untreated and treated mice. Molecular weight is labelled on the right. (**C**) The graph summarizes the results of band density measurements for Wisp1 and GAPDH in the kidneys. Protein values are expressed relative to normal control, which was set to unity. * *p* < 0.05, vs. NC; # *p* < 0.05, vs. db/db–14wk; £ *p* < 0.05, vs. db/db–22wk.

**Table 1 ijms-25-09651-t001:** Clinical parameters of the experimental groups of mice.

	NC–22wk (n = 8)	db/db–14wk (n = 9)	db/db–22wk (n = 9)	db/db + NCS–22wk (n = 9)
BW, initial, g	26.3 ± 1.08	47.3 ± 2.49 *	48.2 ± 2.19 *	48.5 ± 3.55 *
BW, 18wk, g	27.4 ± 1.69		51.1 ± 4.35 *	54.3 ± 3.53 *
BW, final, g	28.3 ± 1.79		47.3 ± 6.75 *	47.4 ± 6.95 *
plasma glucose, initial, mg/dL	128.3 ± 34.7	500.3 ± 109.6 *	504.7 ± 67.0 *	504 ± 77.7 *
plasma glucose, final, mg/dL	161.8 ± 52.8		593.5 ± 18.3 *#$	557.7 ± 45.1 *#£$
HbA1c, initial, %	4.13 ± 0.15	7.77 ± 1.17 *	7.57 ± 1.14 *	7.64 ± 0.87 *
HbA1c, final, %	4.52 ± 0.36		13.35 ± 0.97 *#$	10.87 ± 1.09 *#£$
urine volume, initial, mL/d	0.8 ± 0.67	13.0 ± 5.96 *	14.75 ± 5.72 *	11.06 ± 4.93 *
urine volume, 18wk, mL/d			26.4 ± 5.98 $	24.0 ± 6.05 $
urine volume, final, mL/d	0.67 ± 0.36		35.7 ± 4.65 *&	13.0 ± 2.51 *£&
KW, g	0.238 ± 0.03	0.359 ± 0.04 *	0.509 ± 0.04 *#	0.402 ± 0.04 *£
plasma BUN, mg/dL	20.88 ± 4.73	42.66 ± 3.96 *	45.76 ± 4.13 *	30.98 ± 2.96 *#£
plasma Cr, mg/dL	0.18 ± 0.02	0.31 ± 0.03 *	0.33 ± 0.03 *	0.24 ± 0.02 *#£

Unless specified, parameters were recorded at the end of the experimental period (22 weeks of age). BW, body weight. d, day. KW, kidney weight. Cr, creatinine. * *p* < 0.05 vs. NC; # *p* < 0.05, vs. db/db–14wk; £ *p* < 0.05 vs. db/db–22wk; $ *p* < 0.09, vs. its own initial levels (14 weeks of age) and & *p* < 0.05, vs. its own levels at 18 weeks of age.

**Table 2 ijms-25-09651-t002:** Treatment with NCS reduced urinary inflammatory and oxidative stress markers in db/db mice.

Urine Levels	NC–22wk (n = 8)	db/db–14wk (n = 9)	db/db–22wk (n = 9)	db/db + NCS–22wk (n = 9)
TNF-a, pg/24 h	1.22 ± 0.41	20.21 ± 3.17	88.53 ± 36.73	15.16 ± 1.91
logTNF-a, pg/24 h	−0.004 ± 0.16	1.24 ± 0.09 *	1.76 ± 0.12 *#	1.16 ± 0.05 *£
MCP-1, pg/24 h	11.81 ± 2.42	205.74 ± 31.05	687.97 ± 126.79	148.67 ± 13.07
logMCP-1, pg/24 h	1.04 ± 0.09	2.26 ± 0.07 *	2.79 ± 0.06 *#	2.16 ± 0.04 *£
MDA, µmol/24 h	0.022 ± 0.002	11.56 ± 1.94	37.02 ± 5.62	9.69 ± 1.00
logMDA, µmol/24 h	−1.66 ± 0.04	1.01 ± 0.06 *	1.54 ± 0.4 *#	1.00 ± 0.03 *£

* *p* < 0.05 vs. NC; # *p* < 0.05, vs. db/db–14wk; £ *p* < 0.05 vs. db/db–22wk.

**Table 3 ijms-25-09651-t003:** Primer list for q-PCR.

Gene	Primer	Sequence 5′–3′
mouse	Forward	CCGTGGGATGTTTGAGACTT
FN	Reverse	GGCAAAAGAAAGCAGAGGTG
mouse	Forward	ACGTCCTGGTGAAGTTGGTC
Col-Ia1	Reverse	CAGGGAAGCCTCTTTCTCCT
mouse	Forward	CACCCATCTCTGGGGACAAC
Col-IVa1	Reverse	GTTAGGGCACTGCGGAATCT
mouse	Forward	GTCCAGGACTTCACAATTGAGC
Wisp1	Reverse	CCAGGCTTTGCTTCCATTG
mouse	Forward	ACCCAGAAGACTGTGGATGG
GAPDH	Reverse	CACATTGGGGGTAGGAACAC

## Data Availability

Please contact the corresponding author for data requests. These data were presented in part at the annual Kidney Week of the American Society of Nephrology; 2–5 November 2023.

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
