# Peer review of "Repurposing Niclosamide to Modulate Renal RNA-Binding Protein HuR for the Treatment of Diabetic Nephropathy in db/db Mice"

_ijms, 2024, doi:10.3390/ijms25179651_

Round 1

Reviewer 1 Report

Comments and Suggestions for Authors

The paper entitled “Repurposing niclosamide to modulate renal RNA-binding protein HuR for the treatment of diabetic nephropathy in db/db mice” is an interesting research effort to shed light on the effects of niclosamide in diabetic nephropathy. However, some critical points should be addressed.

The authors should justify the route of administration and the dosage of niclosamide used.

In addition, the applicability of the findings in mice to humans should be further discussed.

The term “in vivo” must be written in italics.

The percentage of plagiarism should be also decreased.

Comments on the Quality of English Language

Minor editing of English language required.

Author Response

Reviewer 1 comments

The paper entitled “Repurposing niclosamide to modulate renal RNA-binding protein HuR for the treatment of diabetic nephropathy in db/db mice” is an interesting research effort to shed light on the effects of niclosamide in diabetic nephropathy. However, some critical points should be addressed.

The authors should justify the route of administration and the dosage of niclosamide used.

Response: yes, we have included the requested information on page 10 under the section of Animals and Experimental Designs.

In addition, the applicability of the findings in mice to humans should be further discussed.

Response: Yes, the applicability of the findings in mice to humans has been further discussed on page 9 under the section of Discussion.

The term “in vivo” must be written in italics.

Response: thanks. It was now written in italics.

The percentage of plagiarism should be also decreased.

Response: Regarding the issue of plagiarism, we take this matter very seriously. While we understand the sensitivity of this concern, we would like to clarify that any similarities, particularly in methods, with our previous publications have been significantly revised. We have made substantial changes to the writing to ensure that the manuscript reflects new and original content.

Reviewer 2 Report

Comments and Suggestions for Authors

The article is well-written and provides a sufficient overview of the topic.

Here are some suggestions:

1.      The Introduction is clear but could benefit from a more direct statement on the limitations of current treatments and the urgency of finding novel therapeutic targets. This would better set up the rationale for the study. More in details, it could benefit from a brief discussion on the current state of research regarding HuR inhibitors, specifically why existing inhibitors like KH3 and KH39 haven't yet been translated into clinical practice. This would underscore the significance of repurposing niclosamide.

2.      You mention that NCS has shown promise in other contexts, but briefly summarizing these successes could better justify your hypothesis that NCS will work in DKD

3.      For the Materials and methods part, it's important to include catalog numbers for key reagents and antibodies.

4.      For the histological analyses, please include details on the image analysis software (e.g., version, manufacturer).

5.      When discussing the effects of NCS treatment, be clear whether it reduces overall HuR expression or affects specific cellular localizations (Figure 1). NSC treatment reduced the total expression or its expression in a specific cellular localisation? (Figure 1). Arrows or markers should clearly indicate areas of interest, such as cytoplasmic versus nuclear localization of HuR.

6.      Please add Masson’s Trichrome staining to complement PAS staining for clarity on fibrosis. Additionally, merging Figures 3 and 4 could help the presentation of data related to kidney hypertrophy and fibrosis.

7.      The discussion is dense with informations. Consider breaking down complex sentences and eliminating redundancy to enhance readability.

8.      While the discussion touches on the therapeutic potential of NCS, it would benefit from a more explicit discussion of the clinical implications, especially considering the long-term safety and potential side effects of NCS in human populations.

9.      The discussion could benefit from a more detailed outline of future research directions. Suggestions could include exploring the long-term effects of NCS treatment in other models of DN, or investigating the potential for combining NCS with other therapeutic agents.

Comments on the Quality of English Language

The English level is appropriate.

Author Response

Reviewer 2

Here are some suggestions:

  1. The Introduction is clear but could benefit from a more direct statement on the limitations of current treatments and the urgency of finding novel therapeutic targets. This would better set up the rationale for the study. More in details, it could benefit from a brief discussion on the current state of research regarding HuR inhibitors, specifically why existing inhibitors like KH3 and KH39 haven't yet been translated into clinical practice. This would underscore the significance of repurposing niclosamide.

Response: Thank reviewer 2 for this suggestion. We have added this suggested information to the introduction on page 3.

  1. You mention that NCS has shown promise in other contexts, but briefly summarizing these successes could better justify your hypothesis that NCS will work in DKD

Response: Thanks again for this suggestion. This information has also added to the introduction on page 4.

  1. For the Materials and methods part, it's important to include catalog numbers for key reagents and antibodies.

Response: Catalog numbers for key reagents have been included in the Materials and methods section. Details on the antibody resource including catalog number and usage information, are provided in Supplementary Table S1.

  1. For the histological analyses, please include details on the image analysis software (e.g., version, manufacturer).

Response: This has been included on page 11.

  1. When discussing the effects of NCS treatment, be clear whether it reduces overall HuR expression or affects specific cellular localizations (Figure 1). NSC treatment reduced the total expression or its expression in a specific cellular localisation? (Figure 1). Arrows or markers should clearly indicate areas of interest, such as cytoplasmic versus nuclear localization of HuR.

 Response: NCS treatment decreased the overall HuR staining and total HuR protein levels in diabetic kidneys. Arrows indicate a few cytoplasmic HuR staining.  More accurate description was included on page 4.

  1. Please add Masson’s Trichrome staining to complement PAS staining for clarity on fibrosis. Additionally, merging Figures 3 and 4 could help the presentation of data related to kidney hypertrophy and fibrosis.

 Response: Thanks for this good suggestion. First, we perform the collagen IV staining and included this new data to further clarify renal fibrosis. We usually asked our university histology core to carry out Masson’s Trichrome staining for us and it will take two to four weeks. We were asked to complete this revision in a week. It is why we conducted the collagen IV immunofluorescent staining not Masson’s Trichrome staining in the revised version.

Second, Figures 3 and 4 were combined.

  1. Thediscussion is dense with informations. Consider breaking down complex sentences and eliminating redundancy to enhance readability.

Response: Thanks a lot. The entire Discussion section has been revised.

  1. While the discussion touches on the therapeutic potential of NCS, it would benefit from a more explicit discussion of the clinical implications, especially considering the long-term safety and potential side effects of NCS in human populations.

Response: Yes, it has been discussed. Please see the revised version in red color.

  1. The discussion could benefit from a more detailed outline of future research directions. Suggestions could include exploring the long-term effects of NCS treatment in other models of DN, or investigating the potential for combining NCS with other therapeutic agents.

Response: yes, the future direction should focus on investigating the potential for combining NCS with other therapeutic agents for the treatment of DN or other CKD.